# ALIGNING DRAFT AND TARGET IN SPECULATIVE DECODING: A COT-AWARE AND SR-GUIDED MIXED FRAMEWORK

## ABSTRACT

Chain-of-thought (CoT) prompting enhances the reasoning of large language models (LLMs) but increases autoregressive latency; speculative decoding (SD) mitigates this via a small-drafter–large-verifier pipeline whose efficiency hinges on drafted-token acceptance. We show that training-based SD methods (e.g., EAGLE) suffer from catastrophic forgetting and distribution shift under naïve CoT supervision, and we propose a CoT-aware mixed training framework that raises acceptance without altering decoding hyperparameters by combining (i) process-level CoT distillation with feature regression to reduce forward KL divergence and improve step-wise acceptance, and (ii) SURPRISERATIO (SR), a data-selection metric that anchors the distribution and prevents forgetting using minimal open-domain samples. A two-stage mixed-training schedule further balances task alignment and generalization.Experiments on two target models show that our methods achieve wall-clock speedups of $3.04\times$–$4.55\times$ across three datasets, while increasing the average acceptance length by$2.76\times$–$5.62\times$.

## 1 INTRODUCTION

Large language models (LLMs) have demonstrated robust performance across a wide range of natural language tasks, including machine translation, summarization, code generation, and dialogue, thereby establishing themselves as versatile tools for both understanding and generation (OpenAI, 2023). Chain-of-thought (CoT) prompting further enhances complex reasoning by eliciting the model to and generate intermediate problem-solving steps, proving particularly effective in mathematical reasoning, logical inference, and multi-step problem solving. However, the autoregressive nature of LLMs introduces significant computational cost and latency (Huang et al., 2025), and the generation of additional tokens via CoT exacerbates this overhead. This bottleneck severely limits the deployment of such models in real-time, large-scale systems where efficiency is critical.

Recent efforts have introduced speculative decoding (SD) algorithms to mitigate the computational and latency costs associated with autoregressive decoding (Hong, 2025). SD emplots a smaller, faster draft model to propose token sequences, which are verified in parallel by a larger target model. Accepted candidate tokens are retained; otherwise, remaining tokens are resampled. As a result, the effectiveness of SD depends directly on the candidate token acceptance rate: higher acceptance yields greater speedup.

Existing SD approaches can be categorized into two groups: training-free methods, which use pretrained models directly as draft models, and training-based methods, which fine-tune or retrain draft models to better align with target model output distributions. Compared to training-free approaches, training-based SD generally achieves better distribution alignment, leading to longer accepted token sequences and higher speed-ups (Gautam et al., 2025).The most straightforward method to adapt SD algorithms such as EAGLE for CoT contexts can be two-step pipeline: pre-training on open-domain corpora followed by fine-tuning with CoT trajectories via supervised fine-tuning (SFT).Unfortunately, this simple approach has proven ineffective in practice, even underperforming the original model.

This finding suggest that merely applying SFT with CoT data to a generally pre-trained draft model is insufficient. Previous work indicates that the fundamental limitation of conventional

SFT in CoT settings is **catastrophic forgetting** (Kotha et al., 2024): task-specific or domain-specific (in-distribution/narrow-domain) alignment tends to degrade the model's general (out-of-distribution/open-domain) capabilities. While mixed-data training offers a direct remedy, the vast scale of open-domain data makes it prohibitively expensive and introduces significant distributional noise when applied broadly across CoT tasks.

Recent studies (Elhady et al., 2025) suggest that continued exposure to a curated subset of open-domain data during narrow-domain learning can effectively suppress **parameter drift** toward a single subdomain, thereby preserving general capabilities and improving generalization. This, in turn, raises a natural question: Can draft models perform self-directed open-domain data mining to acquire the knowledge they lack?

Motivated by this question, we depart from the conventional sequential "pretrain-then-fine-tune" pipeline. Instead, we adopt a training scheme that couples targeted CoT supervision with SR-metric–guided self-mining. This design enables the draft model to acquire missing open-domain knowledge while preserving task-specific competence, thereby adapting SD algorithms to CoT scenarios (Objective I) and mitigating catastrophic forgetting (Objective II). In doing so, it balances task-specific alignment with general-purpose capability.

For *Objective I*, we perform process-level distillation with CoT hard labels, treating EAGLE training as driving the draft distribution toward the target distribution. Using a CoT trigger (e.g., "Let's think step by step"), we offline-generate complete CoT trajectories with the target model, then apply process-supervised training via teacher-forced maximum likelihood estimation (MLE) combined with feature regression. This directly aligns distributions along the trajectory, explicitly reduces per-step Kullback–Leibler(KL) divergence and Total Variation Distance (TVD), and increases the acceptance rate $\beta_t$ and accepted length $\tau$.

For *Objective II*, we introduce a mixed training strategy guided by SurpriseRatio (SR) for open-domain data selection. Recognizing that SD efficiency hinges on draft–target interactions, we first train a draft model on CoT data, then use the draft model to identify missing open-domain knowledge via the SR metric—tokens the large model would accept but the small model has not yet learned. We compute token-level signals $s_t$ on the ShareGPT dataset and apply SR-based adaptability filtering, which lowers supervision cost, preserves broad knowledge, and prevents convergence toward narrow-domain bias.

**Contributions.**

- **Process-distillation analysis and implementation from a distribution-alignment view.** We provide a complete mathematical proof that, in speculative decoding, training the draft model on target-model–distilled labels is sound and effective. We show that teacher-trajectory CoT conditional MLE (with EAGLE) lowers the *forward-KL* under teacher context, thereby shrinking TVD, increasing acceptance $\beta_t$, and supplying an alignment-centric interpretation of SD which is better suited to CoT scenarios.

- **Interaction-driven, SR-guided data selection.** We define the *SurpriseRatio* as a token-to-sample-level adoptability prior. This yields a *lower-cost, supervision-light* way to curate data that contains **richer open-domain knowledge (which the current model lacks)** and is more likely to be accepted by the target, thereby improving SD efficiency.

- **Mixed training scheme and empirical validation.** Instead of a purely serial "general-pretrain then fine-tune" pipeline, we adopt full CoT + SR mixed training. Compared to vanilla EAGLE, our method consistently achieves higher speed-up ratios and longer average accepted lengths across varying temperature settings. Moreover, our approach matches the performance of methods trained on the full mixed open-domain dataset while using significantly less data.

## 2 RELATED WORK

### 2.1 SPECULATIVE DECODING

The primary latency bottleneck in LLM inference arises from the sequential nature of autoregressive decoding, where memory-bound operations dominate (Pope et al., 2022). Speculative decoding (SD) (Leviathan et al., 2022; Chen et al., 2023) mitigates this by introducing a lightweight draft

model to propose multiple tokens per step, which are then verified in parallel by a larger target model. Earlier parallel decoding approaches—such as block-parallel sampling (Stern et al., 2018) and aggressive decoding—struggled with issues like prediction drift, input constraints, or limited stochasticity. Recent variants of SD optimize the interaction between the draft and target models and incorporate tree-structured verification (Sun et al., 2023; Miao et al., 2023; Cai et al., 2024). EAGLE (Li et al., 2024a) further rethinks SD by performing autoregression over hidden features with shifted-token conditioning, improving draft accuracy. Its successors, such as EAGLE-2 and EAGLE-3 (Li et al., 2024b; 2025), introduce dynamic tree structures and multi-layer feature fusion, achieving up to $6.5\times$ speedup without quality loss.

While EAGLE relies on the full ShareGPT corpus, we propose a SurpriseRatio (SR) strategy that combines filtered ShareGPT data with a task-specific dataset. This approach reduces the scale of data required while maintaining—and in some cases even improving—downstream performance. Our results demonstrate that careful task-aware data selection enhances both computational and modeling efficiency.

## 2.2 DATA SELECTION

Selecting high-utility data subsets from large-scale corpora is critical for both general and task-specific large LLM training, with the core objective of improving data efficiency and model performance (Zhao et al., 2023). Existing strategies generally follow two directions: one focuses on enhancing data diversity through clustering and embedding-based sampling (Sorscher et al., 2022; Tirumala et al., 2023; Wu et al., 2023); the other aims to improve data quality via instruction filtering and difficulty estimation (Xu et al., 2023; Wang et al., 2023). Recent research pipelines further integrate both approaches to balance diversity and quality, thereby achieving more efficient model training (Zhou et al., 2023a).

## 2.3 KNOWLEDGE DISTILLATION

Knowledge distillation (KD) (Hinton et al., 2015) trains compact student models under the guidance of larger teacher models, reducing inference cost while preserving performance. Early KD methods for LLMs operated primarily in a black-box setting, relying solely on teacher outputs. The rise of open-source models (Zhang et al., 2022; Touvron et al., 2023) enabled white-box KD, leveraging richer internal signals for distillation. While prior work integrated KD into decoding to improve sampling or acceptance rates, these approaches typically targeted general scenarios rather than tailoring student models for domain-specific reasoning. In this work, we repurpose KD not for model compression, but to enhance SD: task-specific data are distilled using hard labels generated by a large teacher model, then mixed with a filtered open-domain subset for retraining. This hybrid approach significantly improves task alignment and SD efficiency under constrained data and computational budgets.

## 3 PRELIMINARIES

### 3.1 SPECULATIVE DECODING FRAMEWORK

SD accelerates autoregressive language model inference by leveraging a smaller draft model $\mathcal{M}^{(s)}$ to generate candidate tokens, which are then verified in parallel by the target model $\mathcal{M}^{(t)}$. Given context $\rho = \{x, y_{<t}\}$, the process operates in three stages: (1) draft generation of $L$ candidate tokens using $\mathcal{M}^{(s)}$; (2) parallel evaluation by $\mathcal{M}^{(t)}$ to compute acceptance probabilities; (3) rejection sampling to maintain the target distribution while accepting verified tokens.

The key efficiency metric is the acceptance rate, defined as the expected proportion of draft tokens retained per generation step. Higher alignment between the draft and target distributions leads to improved acceptance rates and greater overall speedup.

EAGLE as a notable framework in SD areas constructs a regression objective that aligns draft features with target hidden representations, enabling more accurate candidate generation and higher acceptance rates compared to traditional independent draft models.

## 3.2 CHAIN-OF-THOUGHT UNDER SPECULATIVE DECODING: REASONING, ALIGNMENT, AND FORGETTING

CoT prompting improves complex reasoning by making intermediate steps explicit, but it also lengthens sequences and increases inference costs. In SD setting, these effects intensify: the fast draft model must align with the high-accuracy target not only on final answers but along entire reasoning trajectories to maintain acceptance rates and overall efficiency. The core challenge is the **tension between distributional alignment and forgetting**. Let $q(\cdot)$ denote the draft distribution and $p(\cdot)$ the target distribution. Effective SD requires $q \approx p$ across diverse contexts and over the full sequence of CoT tokens. Yet fine-tuning the draft on task-specific CoT data often over-specializes it to particular reasoning patterns, shifting it away from general-domain behavior and inducing *catastrophic forgetting*. In CoT adaptation, such misalignment compounds along long reasoning chains, lowering acceptance rates, increasing fallback costs, and ultimately reducing SD efficiency.

We therefore pose the objective as maximizing performance subject to alignment and stability constraints. This formulation makes explicit the defining trade-off in CoT under SD—improving speedup ratios versus preserving draft-to-target alignment—and motivates process-token metrics.

## 4 METHODOLOGY

We aim to adapt SD to CoT-centric reasoning while preserving the draft model's general capabilities. Our method couples (1) **process-level CoT distillation** that aligns the draft to the target distribution along teacher trajectories, with (2) **SurpriseRatio-guided open-domain selection** that curates interaction-friendly supervision from large general corpora. We then train with a **mixed scheme** that interleaves both signals under a fixed decoding setup (identical to EAGLE's verifier and search hyperparameters). This section formalizes the objectives, clarifies why they improve token acceptance in SD, and details the training procedure.

### 4.1 PROCESS-LEVEL CoT DISTILLATION

**Setting.** Let the target model be $p(\cdot \mid x_{<t})$ and the draft be $q(\cdot \mid x_{<t})$. We elicit CoT traces with a fixed prompting policy and collect sequences $\mathcal{D}_{\text{CoT}} = \{x_{1:T}\}$ comprising intermediate reasoning and final answers. For each prefix $x_{<t}$ generated by the tearcher model, the draft is supervised using hard labels $x_t$ provided by the teacher, along with EAGLE-style feature regression applied to the hidden states.

**Objective.** The token loss is defined as the conditional NLL averaged over teacher contexts:

$$\mathcal{L}_{\text{NLL}} = \mathbb{E}_{x_{<t}\sim p,\, x_t \sim p(\cdot|x_{<t})}[-\log q(x_t \mid x_{<t})] = \mathbb{E}_{x_{<t}\sim p}[H(p(\cdot \mid x_{<t}),\, q(\cdot \mid x_{<t}))], \quad (1)$$

which, by identity, can be expressed as $\mathbb{E}_{x_{<t}\sim p}[H(p(\cdot \mid x_{<t})) + \text{KL}(p \parallel q)]$. This loss is combined with a feature regression loss between the target and draft hidden states (denoting the target feature as $f_t$ and the draft prediction as $\hat{f}_t$ :

$$\mathcal{L}_{\text{feat}} = \mathbb{E}_{x_{<t}\sim p} \|f_t - \hat{f}_t\|_2^2, \quad (2)$$

which contracts the logit discrepancy when the output layer is linear $z_t = W f_t + b$, since

$$\|z_t - \hat{z}_t\|_2 \leq \|W\| \|f_t - \hat{f}_t\|_2, \quad (3)$$

and stabilizes $q(\cdot \mid x_{<t})$ under the softmax mapping. Equivalently, this contracts the student–teacher gap from two complementary perspectives—features (representation) and probabilities (NLL)—leading to smoother and more controlled forward-KL convergence on teacher CoT prefixes, enlarging the overlap through $\min\{p(\cdot \mid x_{<t}),\, q(\cdot \mid x_{<t})\}$, and thus increasing the acceptance rate.

**Implications for SD acceptance.** In rejection-based SD with verifier $p$, the token-wise acceptance at step $t$ is

$$\beta_t = \sum_x \min\{p(x \mid x_{<t}),\, q(x \mid x_{<t})\} = 1 - \text{TVD}(p(\cdot \mid x_{<t}),\, q(\cdot \mid x_{<t})). \quad (4)$$

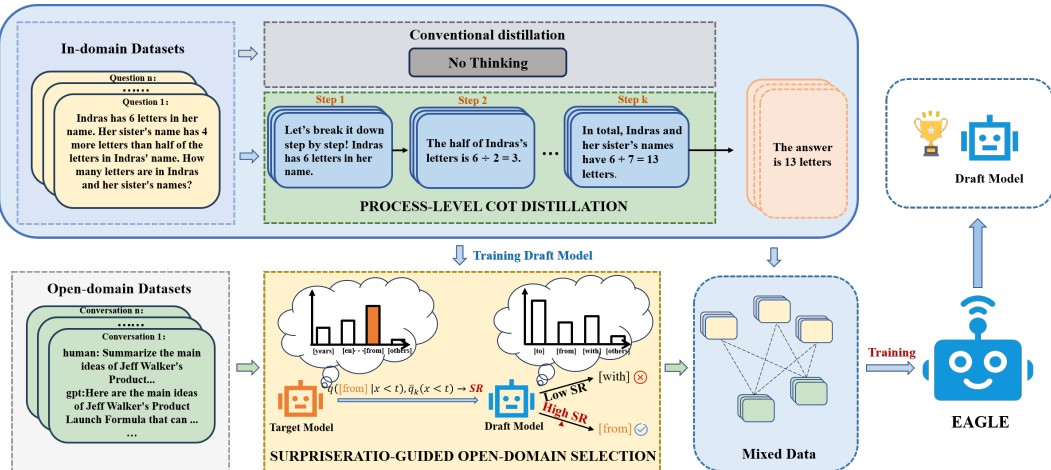

Figure 1: Overall training pipeline of our method. In-domain datasets (e.g., GSM8K, ASDiv) are distilled with **process-level CoT supervision**, contrasting with conventional distillation that only transfers final answers. For open-domain datasets (e.g., ShareGPT), we apply our **SurpriseRatio-guided selection** to select a high-quality subset. The distilled in-domain data and SR-selected open-domain data are then **mixed and jointly trained** under the EAGLE framework, producing an efficient draft model for speculative decoding.

We supervise the draft on **teacher CoT prefixes** using conditional MLE, which equals minimizing the forward KL under teacher contexts via the identity given in equation 1. This reduces the discrepancy between the conditional distributions of the verifier and the draft along every prefix where verification occurs. Pinsker's inequality yields $\mathrm{TVD}(p,q) \leq \sqrt{\frac{1}{2}\,\mathrm{KL}(p \parallel q)}$, shrinking the forward KL on these contexts tightens a Pinsker bound on TVD in equation 4, thereby increasing the expected acceptance rate $\alpha = \mathbb{E}[\beta_t]$. This, in turn, raises the expected accepted length $\tau = \frac{1-\alpha^{r+1}}{1-\alpha}$ for a block size $r$.

Meanwhile, training on step-wise hard labels derived from teacher CoTs exposes the draft model to low-entropy, process-structured distributions ("First..., Then..., Finally..."), providing explicit intermediate guidance that better aligns the draft with the verifier's probability mass in regions where actual reasoning occurs.

**Takeway.** Long CoT sequences are prone to error accumulation. Step-wise supervision localizes credit assignment: by ensuring that forward $\mathrm{KL}(p \parallel q)$ does not increase—preferably decreases—on each teacher prefix, the average TVD is reduced via Pinsker's inequality, which monotonically increases $\alpha$ and hence $\tau$.

### 4.2 SURPRISERATIO-GUIDED OPEN-DOMAIN SELECTION

**Motivation.** Training a draft model solely on pure CoT supervision risks over-specialization and poor generalization. To mitigate this challenge, a natural fix is to incorporate broad conversational corpora such as ShareGPT, but naïvely ingesting all such data is inefficient and increases the cost of adapting to new CoT domains. This motivates the question of whether curated general-corpus data, when mixed with dedicated CoT data, can more effectively train the draft model. So in practice, we propose a targeted alternative: first pretrain the draft model on a specialized CoT set, then allow the model to guide the retrieval of examples from a general corpus, selecting only those predicted to best compensate for its current weaknesses.

Concretely, we introduce **SR**, a token-level agreement metric that compares the draft model's surprisal with that of a target model on the same continuation. Intuitively, when the target assigns high probability to a token, the draft should also do the same, SR quantifies this alignment.

**Definition.** For a prefix $x_{<t}$ and draft distribution $q$, let $\mathcal{N}_k(x_{<t})$ be the top-$k$ candidate set under $q(\cdot \mid x_{<t})$ and define $\bar{q}_k(x_{<t}) = \frac{1}{k}\sum_{x \in \mathcal{N}_k(x_{<t})} q(x \mid x_{<t})$. Given a supervised token $x_t$ (obtained from teacher CoT traces or high-quality references) **that the target model is likely to accept, the** *SurpriseRatio* **at step** $t$ **is defined as:**

$$\mathrm{SR}(x_t; x_{<t}) = \frac{q(x_t \mid x_{<t})}{\bar{q}_k(x_{<t})}. \tag{5}$$

When SR is high, the draft model demonstrates a strong preference for the supervised token compared to local alternatives. This increases the likelihood that $q(\bullet \mid x_{<t})$ overlaps with the high-probability region of the verifier's distribution, leading to higher acceptance according to Eq. equation 4. A high SR arises when $q(x_t \mid x_{<t})$ is large—indicating the draft model strongly favors the teacher-provided token—and the $\bar{q}_k(x_{<t})$ is small—indicating that the token already occupies substantial probability mass in the draft model's distribution. Under SD, this alignment narrows the gap between the draft and target distributions, thereby increasing token acceptance probability.

**A sequence score aggregates token scores, e.g.,**

$$S(x_{1:T}) = \frac{1}{T}\sum_{t=1}^{T} \mathbf{1}\left[\,\mathrm{SR}(x_t; x_{<t}) > \tau_{\mathrm{SR}}\,\right]. \tag{6}$$

Selecting samples with high $S(x_{1:T})$ thus prioritizes data that (i) reinforce regions where the draft and target already interact smoothly, and (ii) encourage longer accepted spans under SD. In addition, we enable the draft model to mine large open-domain corpora for information that may be under-represented due to CoT task-specific supervision: by comparing its open-domain confidence to its supervised preferences, the draft identifies tokens and states where its coverage is sparse or biases are over-amplified, such cases are then up-weighted during selection.

**Practical computation.** We compute SR on open-domain logs by: (1) running the small draft to obtain $q(\cdot \mid x_{<t})$, (2) extracting top-$k$ mass and $q(x_t \mid x_{<t})$ for the provided continuation token $x_t$ (from the log), and (3) retaining sequences whose $S(x_{1:T})$ exceeds a threshold. **Thresholds and** $k$ **are tuned once on a held-out split and then fixed.(Not clear)**

### 4.3 MIXED TRAINING SCHEME

**Dataset composition.** Let $\mathcal{D}_{\mathrm{SR}}$ be the SR-filtered subset from ShareGPT and $\mathcal{D}_{\mathrm{rand}}$ a small random subset (regularizer). Training is performed on the mixed dataset:

$$\mathcal{D}_{\mathrm{mix}} = \mathcal{D}_{\mathrm{CoT}} \cup \mathcal{D}_{\mathrm{SR}} \cup \mathcal{D}_{\mathrm{rand}},$$

with sampling ratios $(1 : \lambda : \varepsilon)$, where $\lambda$ controls the open-domain proportion and $\varepsilon \ll \lambda$ helps prevent overfitting to SR-selected modes.

**Loss and schedule.** The final objective is

$$\mathcal{L} = \lambda' \mathcal{L}_{\mathrm{NLL}}(\mathcal{D}_{\mathrm{mix}}) + \mu' \mathcal{L}_{\mathrm{feat}}, \tag{7}$$

with fixed verifier/decoding hyperparameters inherited from EAGLE. We adopt a simple two-stage schedule found to be robust: first warm up on $\mathcal{D}_{\mathrm{CoT}}$ to stabilize feature autoregression, then incorporate $\mathcal{D}_{\mathrm{SR}}$ at ratio $\lambda$ while maintaining a small proportion $\varepsilon$ of random samples. This strategy preserves a distributional anchor and mitigates catastrophic forgetting without requiring verifier re-training.

**Discussion.** The framework is complementary to other KD/SD refinements (e.g., divergence (Zhou et al., 2023b)- or ranking-aware losses, harmonized context alignment (Zhang et al., 2024)). These can be layered on top of our objectives without altering the mixed data policy, thereby offering a path to further improvements while preserving the core acceptance rationale.

# 5 EXPERIMENT

## 5.1 EXPERIMENTAL SETUP

We evaluate training strategies for EAGLE drafts under **lossless** speculative decoding. We maintain the same parameter Settings as eagle, targets are **frozen** with NVIDIA RTX4090D GPU, and we report **efficiency only**: **wall-time speedup** and **average acceptance length** $\tau$.

**Targets.** Two frozen target LLMs: **Vicuna-7B** and **Llama-3-8B-Instruct**. Targets are used (i) to generate CoT hard labels offline and (ii) as verifiers during speculative decoding. No fine-tuning is applied to targets. **Prompt sets.** Efficiency is measured on **GSM8K**, **AQuA-RAT**, and **ASDiv** problem sets used as prompts. Prompt templates and answer extractors are in Appendix B.

**Training corpora (data policy).**

- *CoT hard labels (full).* For each target, we build a full corpus of teacher-generated CoT traces (intermediate reasoning + final answer) under a fixed decoding policy; whenever CoT supervision is used, this full corpus is included.

- *Open-domain (ShareGPT, partial).* We construct a SurpriseRatio-selected subset to complement CoT supervision. For control experiments we also use a random-matched subset with identical size/domain mix. SR hyperparameters (threshold, top-$k$) are tuned once on a small held-out split and then fixed across datasets.

**Metrics.**

- **Speedup Ratio $\alpha$:** The actual test speedup ratio relative to vanilla auto-regressive decoding.

- **Average acceptance length** $\tau$ (expected number of consecutive draft tokens accepted per verification step).

**Scope of comparisons.** Main comparisons focus on training strategies for EAGLE under a mixed data policy (full CoT + partial ShareGPT).Vanilla EAGLE where the draft model is trained exclusively on the complete ShareGPT and we simultaneously process the data distillation strategy and the data selection strategy. Cross-paper KD variants are not part of the core comparison to avoid data/policy confounds.

Table 1: Speedup Ratios of Ours vs. EAGLE on GSM8K, ASDiv, and Aqua-RAT under Different Temperatures.

| Model | Method | Temperature = 0 | | | | Temperature = 1 | | | |
|---|---|---|---|---|---|---|---|---|---|
| | | GSM8K | ASDiv | Aqua | Mean | GSM8K | ASDiv | Aqua | Mean |
| Llama3 8B | EAGLE | 2.88X | 3.01X | 3.02X | 2.97X | 2.53X | 2.69X | 2.77X | 2.66X |
| | CoT Distill | 3.94X | 3.59X | 3.83X | 3.79X | 3.46X | 3.33X | 3.48X | 3.42X |
| | Gold Answer | 2.47X | 0.82X | 2.48X | 1.92X | 2.34X | 0.80X | 2.30X | 1.81X |
| | Ours | 4.05X | 3.83X | 3.92X | 3.93X | 3.67X | 3.51X | 3.58X | 3.59X |
| Vicuna 7B | EAGLE | 3.49X | 3.46X | 3.62X | 3.52X | 2.96X | 2.77X | 2.93X | 2.88X |
| | CoT Distill | 3.72X | 2.76X | 4.53X | 3.67X | 2.99X | 1.86X | 3.10X | 2.65X |
| | Gold Answer | 3.15X | 1.03X | 3.74X | 2.64X | 2.43X | 0.96X | 2.95X | 2.78X |
| | Ours | 4.11X | 3.54X | 4.55X | 4.06X | 3.07X | 2.79X | 3.26X | 3.04X |

## 5.2 ABLATION

We further analyze the contribution of supervision granularity, SurpriseRatio-based data selection, and robustness to seeds and target models, under the same decoding setup as described in Sections 5.1.

**Supervision granularity.** We compare three levels of supervision: ground-truth outcomes, open-domain Corpus, and template-based CoT traces. As we can see in Figure 2,ground-truth outcomes

serve as a lower bound, since EAGLE decoding is lossless with respect to the frozen targets.By contrast, process-level supervision with teacher CoT traces consistently improves both efficiency metrics, indicating that explicit trajectory supervision stabilizes draft acceptance and enhances data efficiency.

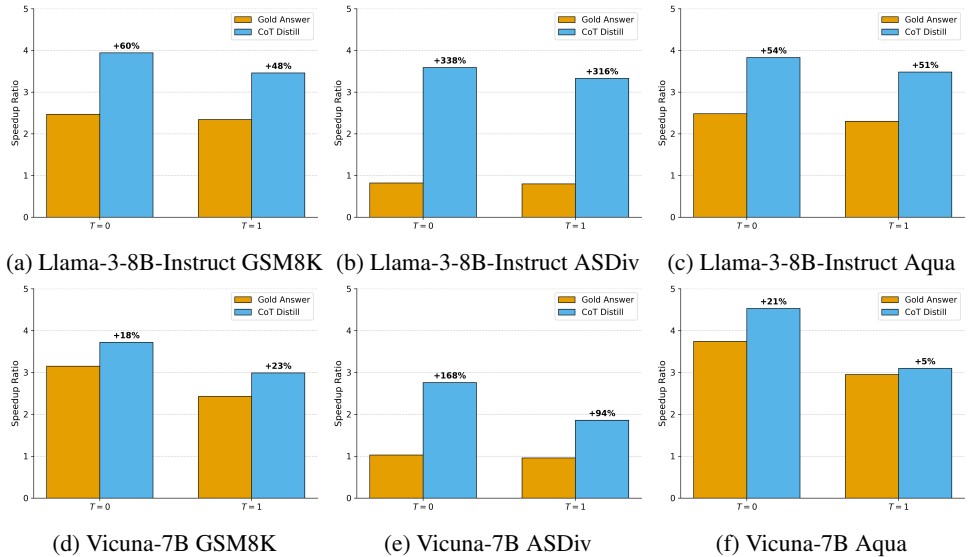

Figure 2: Effect of Supervision Granularity: Process-Level Chain-of-Thought Distillation Yields the Largest Efficiency Gains

**SurpriseRatio selection.** We compare SR-selected subsets against size-matched random baselines and we also use some common metric such as PPL as a reference. Results show that SR filtering yields longer acceptance spans, while achieving higher speedUp ratio. This suggests that targeted curation reduces redundancy without sacrificing effectiveness. In all cases, SR-selected models outperform random baselines, and their acceptance lengths remain stable across GSM8K, AQuA-RAT, and ASDiv prompts.

The improvements therefore cannot be attributed to a specific seed or dataset, but reflect systematic benefits of combining process-level supervision with selective data filtering.

Table 2: Comparison of metrics between Ours and baselines.

| Metric | Speedup Ratio | Acceptance Length | Data Scale |
|---|---|---|---|
| EAGLE | 3.02X | 4.17 | 650G |
| 1% ShareGPT(Perplexity) | 1.72X | 2.22 | 7G |
| 1% ShareGPT(SurpriseRatio) | 1.90X | 2.56 | 7G |
| 1% ShareGPT(Random) | 1.66X | 2.32 | 7G |
| GSM8K + 1% ShareGPT(SurpriseRatio) | 3.00X | 4.27 | 73G |
| GSM8K + 1% ShareGPT(Perplexity) | 2.83X | 3.97 | 73G |
| GSM8K + 1% ShareGPT(Random) | 2.91X | 4.13 | 73G |
| ALPACA + 1% ShareGPT(SurpriseRatio) | 2.83X | 3.22 | 100G |
| ALPACA + 1% ShareGPT(Random) | 2.71X | 3.11 | 100G |

## 5.3 EFFECTIVENESS AND DATA EFFICIENCY

Firstly, we examine the role of SurpriseRatio-based filtering when combining open-domain ShareGPT data with teacher traces.By varying the retention ratio $r$, the result is shown in Figure 2,even incorporating a very small proportion (e.g., 1–5%) of SR-selected ShareGPT data into the in-domain CoT distillation corpus yields substantial gains in both speedup ratio and accepted

length, while further increases bring diminishing returns. Consequently, in our main experiments we report results at the optimal trade-off point, demonstrating that strong performance can be achieved without relying on large amounts of open-domain supervision.

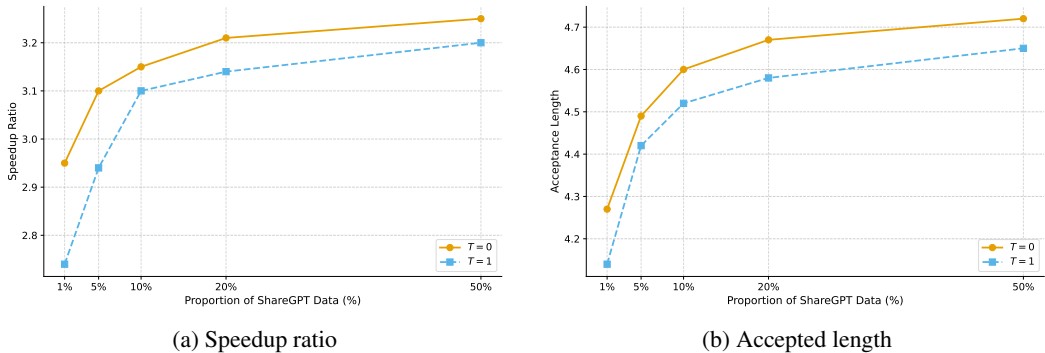

(a) Speedup ratio          (b) Accepted length

Figure 3: Performance of draft models on GSM8K when mixing in different proportions of ShareGPT data selected by our SurpriseRatio (SR) method. (a) Speedup ratio improves significantly with only a small proportion (e.g., 1%–5%), but saturates as more data are included. (b) Accepted length exhibits a similar trend, with early gains followed by diminishing returns.

We next evaluate the effectiveness and data efficiency of different training strategies for EAGLE drafts, addressing two key questions: (i) can a draft trained on a smaller, task-specific corpus (full CoT + partial SR-selected ShareGPT) match or surpass a draft trained on the full ShareGPT? and (ii) within task-specific supervision, does process-level distillation (CoT) outperform outcome-only distillation (answers)? We can see the result in sec5.1 across all datasets (GSM8K, AQuA-RAT, ASDiv), ours achieves comparable or superior speedup relative to vanilla eagle while using a smaller training corpus in terms of data size (GB). This demonstrates that task-specific CoT supervision can effectively replace a large portion of generic open-domain data when training an EAGLE draft,highlighting its improved data efficiency.

Table 3: Acceptance length of Ours vs. EAGLE on GSM8K, ASDiv, and Aqua-RAT under Different Temperatures.

| Model | Method | Temperature = 0 | | | | Temperature = 1 | | | |
|---|---|---|---|---|---|---|---|---|---|
| | | GSM8K | ASDiv | Aqua | Mean | GSM8K | ASDiv | Aqua | Mean |
| Llama3 8B | EAGLE | 4.22 | 4.19 | 4.28 | 4.23 | 4.00 | 4.08 | 4.13 | 4.07 |
| | CoT Distill | 6.03 | 5.36 | 5.48 | 5.62 | 5.63 | 5.13 | 5.20 | 5.32 |
| | Gold Answer | 3.59 | 1.16 | 3.52 | 2.76 | 3.46 | 1.15 | 3.44 | 2.68 |
| | Ours | 5.72 | 5.63 | 5.60 | 2.76 | 5.56 | 5.44 | 5.39 | 5.46 |
| Vicuna 7B | EAGLE | 5.22 | 5.17 | 5.42 | 5.27 | 4.24 | 4.15 | 4.21 | 4.20 |
| | CoT Distill | 5.54 | 4.15 | 5.81 | 5.16 | 4.33 | 3.06 | 4.27 | 3.88 |
| | Gold Answer | 4.38 | 1.45 | 5.49 | 3.77 | 3.53 | 1.35 | 4.32 | 3.07 |
| | Ours | 5.73 | 5.33 | 5.81 | 5.62 | 4.27 | 4.13 | 5.02 | 4.47 |

## 6 CONCLUSION

We introduce a CoT-aware, SR-guided mixed-training framework for SD. It aligns the drafter with the target via process-level CoT distillation and uses SR-based data selection to focus supervision where divergence is highest. On GSM8K, AQUA-RAT, and ASDIV with Vicuna-7B and Llama-3-8B-Instruct verifiers, it delivers **3.04×–4.55×** wall-clock speedups and longer accepted spans than EAGLE; ablations attribute the gains to process-level supervision and SR-guided curation, with only **1–5%** SR-selected data capturing most of the improvement.

## ETHICS STATEMENT

This work adheres to the ICLR Code of Ethics. No human-subject or user data are involved, and therefore no IRB/ethics review was required. We use only public datasets—GSM8K, ASDiv, and Aqua-RAT—which contain no personally identifiable information (PII). We assess the privacy and security risks of our study to be minimal because all data are public and non-sensitive, and our release does not introduce new attack surfaces. We do not perform a fairness/bias evaluation as our experiments do not involve sensitive attributes or demographic subgroups; we instead focus on mathematical reasoning benchmarks. The intended use of our artifacts is *research only*; downstream deployment must account for potential misuse risks unrelated to our setting. We employ pretrained models *Llama 3 8B Instruct* and *Vicuna 7B*; our usage complies with their upstream licenses. We release our code and trained weights for research purposes (see the Reproducibility Statement) and impose no terms beyond the upstream model/data licenses. The authors declare no conflicts of interest.

## REPRODUCIBILITY STATEMENT

Our implementation is built upon the original **EAGLE** codebase (`https://github.com/SafeAILab/EAGLE/`). On top of EAGLE, we refactor the training pipeline to implement our proposed training strategies and restructure the evaluation/testing suite accordingly. We will provide an anonymous repository containing the full codebase, configuration files, and scripts to reproduce all results: (`https://anonymous.4open.science/r/CoT_SR-DF00`). The repository includes an `environment.yml` for exact dependency pinning.

Data used in this paper are public (GSM8K, ASDiv, Aqua-RAT); preprocessing scripts are provided under `ge_data/`. We also include the hard-label generation utilities (paths documented in the repository) to fully reproduce label creation. Training and evaluation follow the original *EAGLE* settings, with our refactored training files and evaluation scripts enabling one-click reproduction of all tables and figures.

Experiments were run on an `RTX 4090D` GPU; seeds, run commands, and configuration files are provided in the anonymous repository. There are no components that we are unable to release.

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

# A APPENDIX

## APPENDIX CONTENTS

## DISCLOSURE OF LLM USAGE

During manuscript preparation, we used a general-purpose large language model tool only for minor copyediting (instruction-based rewriting for grammar and clarity). The tool was not used to generate scientific content, results, code, or citations. All technical statements, method designs, and empirical conclusions were produced by the authors and verified by us; we take full responsibility for the content. No confidential, unpublished, or review-related materials were shared with the tool.

## A.1 PROCESS-LEVEL CoT DISTILLATION (FOR §4.1)

This subsection provides full derivations underlying the theory in §4.1 of the main text, including the NLL–forward-$D_{\mathrm{KL}}$ equivalence, the link between acceptance rate and total variation distance, Pinsker-type propagation to acceptance, and how feature regression controls post-softmax distributional perturbations. We start with notation and the training objective.

**Notation and setup.** Let $D_{\mathrm{CoT}}$ denote CoT trajectories generated by the teacher/verifier $p$. At step $t$, define the conditional distributions under the teacher prefix $x_{<t}$:

$$p_t(\cdot) := p(\cdot \mid x_{<t}), \qquad q_t(\cdot) := q(\cdot \mid x_{<t}),$$

for teacher and student, respectively. Let $f_t$ be the teacher's intermediate feature and $\hat{f}_t$ the student's regressed feature.

**Training objective.** We consider conditional negative log-likelihood (NLL) and a feature-regression (EAGLE-style) loss:

$$\mathcal{L}_{\mathrm{NLL}} = \mathbb{E}_{x_{<t}\sim p,\ x_t\sim p_t}\big[-\log q(x_t \mid x_{<t})\big] = \mathbb{E}_{x_{<t}\sim p}\big[H(p_t, q_t)\big], \tag{8}$$

$$\mathcal{L}_{\mathrm{feat}} = \mathbb{E}_{x_{<t}\sim p}\big\| f_t - \hat{f}_t \big\|_2^2. \tag{9}$$

The total objective is

$$\mathcal{L} = \lambda'\,\mathcal{L}_{\mathrm{NLL}} + \mu'\,\mathcal{L}_{\mathrm{feat}}. \tag{10}$$

[NLL equals forward $D_{\mathrm{KL}}$] For any fixed prefix $x_{<t}$, $H(p_t, q_t) = H(p_t) + D_{\mathrm{KL}}(p_t\|q_t)$. Therefore,

$$\mathcal{L}_{\mathrm{NLL}} = \mathbb{E}_{x_{<t}\sim p}\big[H(p_t) + D_{\mathrm{KL}}(p_t\|q_t)\big]. \tag{11}$$

In particular, $\mathcal{L}_{\mathrm{NLL}}$ attains its minimum in $q_t$ iff $q_t^\star = p_t$ (almost surely w.r.t. $x_{<t} \sim p$).

Use the identity $H(p_t, q_t) = H(p_t) + D_{\mathrm{KL}}(p_t\|q_t)$, then take expectation over $x_{<t} \sim p$ to obtain equation 11. Convexity and Gibbs' inequality yield the unique minimizer $q_t = p_t$.

[Per-step acceptance rate] In the SD rejection-sampling framework, the token-level acceptance rate at step $t$ is

$$\beta_t := \sum_x \min\{p_t(x),\ q_t(x)\}. \tag{12}$$

Let $\alpha := \mathbb{E}[\beta_t]$ be its expectation over steps.

[Acceptance rate equals one minus TVD] Let $(p_t, q_t) := \frac{1}{2}\sum_x |p_t(x) - q_t(x)|$. Then

$$\beta_t = 1 - (p_t, q_t). \tag{13}$$

Using $\min\{a, b\} = \frac{a+b-|a-b|}{2}$ and $\sum_x p_t(x) = \sum_x q_t(x) = 1$,

$$\beta_t = \tfrac{1}{2}\sum_x \big(p_t(x) + q_t(x) - |p_t(x) - q_t(x)|\big) = 1 - \tfrac{1}{2}\sum_x |p_t(x) - q_t(x)| = 1 - (p_t, q_t).$$

[From $D_{\mathrm{KL}}$ to acceptance via Pinsker] For each $t$, Pinsker's inequality yields $(p_t, q_t) \leq \sqrt{\frac{1}{2} D_{\mathrm{KL}}(p_t \| q_t)}$. Combining with section A.1 and taking expectation over $t$, we get

$$\alpha = \mathbb{E}[\beta_t] \ \geq \ 1 - \sqrt{\tfrac{1}{2} \, \mathbb{E}\big[D_{\mathrm{KL}}(p_t \| q_t)\big]} \ . \tag{14}$$

Apply Pinsker and Jensen (since $\sqrt{\cdot}$ is concave): $\mathbb{E}[] \leq \mathbb{E}\big[\sqrt{\frac{1}{2} D_{\mathrm{KL}}}\big] \leq \sqrt{\frac{1}{2} \mathbb{E}[D_{\mathrm{KL}}]}$. Combine with $\beta_t = 1-$ to obtain equation 14.

[Expected accepted streak vs. block size] Suppose each draft block proposes $r$ tokens, which are then verified sequentially, and approximate per-step acceptance as independent with $\mathbb{E}[\beta_t] = \alpha$. Then the expected number of consecutively accepted tokens in one verification is

$$\tau \ = \ \sum_{i=0}^{r} \alpha^{\,i} \ = \ \frac{1 - \alpha^{\,r+1}}{1 - \alpha}, \tag{15}$$

and $\tau$ is strictly increasing in $\alpha$ for $0 \leq \alpha < 1$.

Geometric series summation. Monotonicity follows from $\partial \tau / \partial \alpha > 0$ for $0 \leq \alpha < 1$.

[Linear output head and softmax] Assume a linear output head $z_t = W f_t + b$ for the teacher, and $\hat{z}_t = W \hat{f}_t + b$ for the student; let $\sigma(\cdot)$ denote the softmax.

[Logits contraction induced by feature regression] Under section A.1,

$$z_t - \hat{z}_{t2} \ \leq \ W_2 \, f_t - \hat{f}_{t2}. \tag{16}$$

Immediate from the operator norm definition and Cauchy–Schwarz.

[From feature error to distributional perturbation] The softmax Jacobian $J = \mathrm{Diag}(p) - p p^\top$ satisfies $J_2 \leq \frac{1}{2}$. Under section A.1 and section A.1,

$$\sigma(z_t) - \sigma(\hat{z}_t)_2 \leq \tfrac{1}{2} \, z_t - \hat{z}_{t2} \ \leq \ \tfrac{1}{2} \, W_2 \, f_t - \hat{f}_{t2}, \tag{17}$$
$$\big(p_t, q_t\big) = \tfrac{1}{2} \sigma(z_t) - \sigma(\hat{z}_t)_1 \ \leq \ \tfrac{1}{4} \, \sqrt{V} \, W_2 \, f_t - \hat{f}_{t2}, \tag{18}$$

where $V$ is the vocabulary size.

A Lipschitz upper bound for softmax on the probability simplex follows from $J_2 \leq \frac{1}{2}$; combine with section A.1 to get equation 17. Then apply $\cdot_1 \leq \sqrt{V} \cdot_2$ to obtain equation 18.

[Two-channel convergence: probabilities and features] Together, equation 11 and equation 18 imply: minimizing $\mathcal{L}_{\mathrm{NLL}}$ directly shrinks $D_{\mathrm{KL}}(p_t \| q_t)$, which, by section A.1, raises the average acceptance $\alpha$; in parallel, minimizing $\mathcal{L}_{\mathrm{feat}}$ controls an upper bound on $(p_t, q_t)$, indirectly improving $\beta_t$ and $\tau$ (see section A.1).

[Why forward $D_{\mathrm{KL}}$?] Conditional MLE on the teacher context naturally corresponds to $D_{\mathrm{KL}}(p \| q)$, encouraging the student to cover the teacher's high-probability mass and thus to maximize $\sum_x \min\{p, q\}$ (section A.1). In contrast, reverse $D_{\mathrm{KL}}(q \| p)$ is mode-seeking rather than coverage-oriented.

**Summary.** section A.1 establishes the "NLL = forward $D_{\mathrm{KL}}$" equivalence; section A.1 connect "$D_{\mathrm{KL}} \downarrow \Rightarrow \downarrow \Rightarrow \alpha \uparrow$"; section A.1 characterizes the expected accepted streak $\tau$ versus block size $r$; and section A.1 shows how feature regression, via the linear head and softmax Lipschitzness, controls distributional perturbations. Consequently, process-level supervision locally corrects errors along long CoT trajectories, curbing error accumulation and improving SD throughput and accepted-span length.

A.2 CATASTROPHIC FORGETTING: TOKEN-LEVEL EVIDENCE AND THEORY

This subsection elaborates on the catastrophic forgetting phenomenon discussed in the main text, with token-level evidence and a simple theoretical account that connects distribution shift, forward-$D_{\mathrm{KL}}$, and the SD acceptance/rejection dynamics defined in §A.1.

**Setup.** Let $P$ denote a *general/open-domain* distribution (e.g., ShareGPT-like) and $M$ denote a *math-reasoning* distribution (e.g., GSM8K-style). We train two student models:

- $q^{\mathrm{gen}}$: fine-tuned on $P$ only;
- $q^{\mathrm{math}}$: fine-tuned on $M$ only.

At step $t$ under teacher prefix $x_{<t}$, define $p_t(\cdot)$ as the teacher conditional and $q_t(\cdot)$ as the student conditional. Following §A.1, the per-step acceptance mass is $\beta_t = \sum_x \min\{p_t(x), q_t(x)\} = 1 - (p_t, q_t)$. For a token group $G \subseteq \mathcal{V}$ (e.g., math symbols or function words), define the *group acceptance mass*

$$\beta_t(G) := \sum_{x \in G} \min\{p_t(x), q_t(x)\}, \quad \text{and the group rejection mass} \quad \rho_t(G) := \sum_{x \in G} \big[q_t(x) - \min\{p_t(x), q_t(x)\}\big].$$
(19)

By construction, $\sum_G \rho_t(G) = 1 - \beta_t$, and $\rho_t(G)$ measures how much of the student's probability mass in $G$ is *not* aligned with the teacher at step $t$.

**Empirical observations.** We evaluate token-level rejections in SD verification:

1. When testing $q^{\text{gen}}$ on math reasoning (GSM8K), math-centric tokens (e.g., ", "=", numerals, "find", "Let") exhibit elevated rejection counts; see **????**.

2. Symmetrically, when testing $q^{\text{math}}$ on general scenarios, common linguistic structures (e.g., "the", "of", sentence punctuation) show higher rejection counts; see **??** and the token cloud in **??**.

Both effects are consistent with task-specific fine-tuning harming performance on out-of-domain knowledge (classical catastrophic forgetting).

**Why fine-tuning on $M$ (or $P$) forgets $P$ (or $M$)?** Recall from §A.1 that minimizing conditional NLL equals minimizing the forward-$D_{\text{KL}}$: $\mathcal{L}_{\text{NLL}} = \mathbb{E}_{x_{<t} \sim \mathcal{D}}[H(p_t) + D_{\text{KL}}(p_t \| q_t)]$ for the training distribution $\mathcal{D}$. If we optimize only on $M$, we specifically reduce $\mathbb{E}_{x_{<t} \sim M}[D_{\text{KL}}(p_t \| q_t)]$, but the evaluation criterion on $P$ depends on $\mathbb{E}_{x_{<t} \sim P}[D_{\text{KL}}(p_t \| q_t)]$. Under distribution shift $P \neq M$, the latter need not decrease; indeed, it can *increase*.

A useful decomposition is the importance-weighting identity:

$$\mathbb{E}_{x_{<t} \sim P}\big[D_{\text{KL}}(p_t \| q_t)\big] = \mathbb{E}_{x_{<t} \sim M}[w(x_{<t}) \, D_{\text{KL}}(p_t \| q_t)], \qquad w(x_{<t}) = \frac{dP}{dM}(x_{<t}).$$
(20)

Pure $M$-risk minimization ignores the weights $w$; if the updates preferentially reduce $D_{\text{KL}}$ where $w$ is *small* and leave (or even increase) $D_{\text{KL}}$ where $w$ is *large*, then the $P$-risk can worsen.

[First-order forgetting via gradient conflict] Let $F_{\mathcal{D}}(\theta) := \mathbb{E}_{x_{<t} \sim \mathcal{D}}\big[D_{\text{KL}}(p_t \| q_{t,\theta})\big]$. A gradient step $\theta' = \theta - \eta \nabla F_M(\theta)$ induces the first-order change $F_P(\theta') \approx F_P(\theta) - \eta \langle \nabla F_P(\theta), \nabla F_M(\theta) \rangle$. If $\langle \nabla F_P, \nabla F_M \rangle < 0$ (negative cosine similarity), then $F_P$ *increases*, i.e., performance on $P$ degrades to first order.

First-order Taylor expansion of $F_P$ around $\theta$.

**From increased $D_{\text{KL}}$ to token-level rejections.** By Pinsker and $\beta_t = 1 - (p_t, q_t)$ (see §A.1),

$$(p_t, q_t) \leq \sqrt{\tfrac{1}{2} D_{\text{KL}}(p_t \| q_t)} \quad \Longrightarrow \quad \beta_t \geq 1 - \sqrt{\tfrac{1}{2} D_{\text{KL}}(p_t \| q_t)}.$$
(21)

Hence any increase in $\mathbb{E}_{x_{<t} \sim P}[D_{\text{KL}}(p_t \| q_t)]$ (or in $\mathbb{E}_{x_{<t} \sim M}[\cdot]$) leads to a *decrease* in expected acceptance on that domain and, correspondingly, an *increase* in rejection mass. Specializing equation 19 to a token group $G$,

$$\rho_t(G) = \sum_{x \in G} \big[q_t(x) - \min\{p_t(x), q_t(x)\}\big] = \sum_{x \in G} \max\{q_t(x) - p_t(x), 0\} \uparrow \text{ as } (p_t, q_t) \uparrow,$$
(22)

so groups where $q_t$ over-allocates mass relative to $p_t$ become rejection hotspots. For $q^{\text{gen}}$ evaluated on math, these hotspots concentrate on math symbols/numerals; for $q^{\text{math}}$ on general, they shift to function words and punctuation—precisely what the plots show.

**A measurement recipe (reproducibility).** To quantify forgetting in practice:

1. Partition the vocabulary into semantically meaningful groups $G_1, \ldots, G_K$ (e.g., math symbols, numerals, function words, punctuation).

2. For each step $t$ in evaluation traces, accumulate per-token rejections; aggregate to group rejection counts $R(G_k) = \sum_t \sum_{x \in G_k} \mathbf{1}\{\text{candidate } x \text{ rejected at } t\}$.

3. Normalize either by candidate frequency or by teacher mass in group $G_k$ to obtain comparable rates.

4. Compare $R(G_k)$ (or rates) across $q^{\text{gen}}$ and $q^{\text{math}}$; use paired tests (e.g., Wilcoxon) and report effect sizes.

**Takeaways.** Task-specific fine-tuning reduces forward-$D_{\text{KL}}$ on the training domain but can *increase* it on other domains due to gradient conflict and distribution shift (section A.2, equation 20). Via the $D_{\text{KL}} \longrightarrow \beta$ chain (equation 21), this manifests as lower acceptance and higher rejection mass, concentrated on domain-specific token groups (equation 22). Our token-level analyses and figures provide concrete evidence of this mechanism on GSM8K (math) versus general-language scenarios.

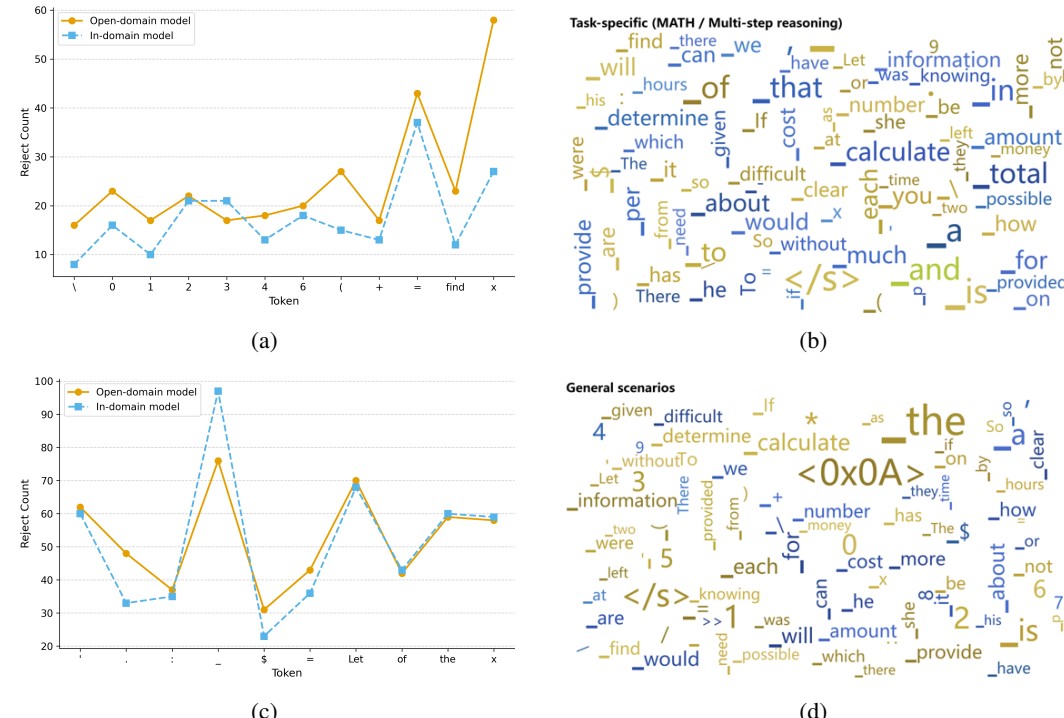

Figure 4: The analysis of most reject tokens which are counted by draft models trained in different scenarios

