# OpenReview forum: "Aligning Draft and Target in Speculative Decoding: A CoT-Aware and SR-Guided Mixed Framework"
_ICLR.cc/2026/Conference — Submitted to ICLR 2026_

### Official Review · Reviewer_MPAw · 2025-10-20

**Soundness:** 2
**Presentation:** 1
**Contribution:** 2
**Rating:** 2
**Confidence:** 4

**Summary:**

The paper SD for CoT prompts and argues that naively fine-tuning draft models on CoT traces causes catastrophic forgetting and lowers token acceptance. It proposes a mixed training framework with two components: process-level CoT distillation that supervises the draft on teacher CoT trajectories with conditional MLE plus feature-regression in an EAGLE-style setup, justified via forward-KL minimization and Pinsker bounds to raise acceptance; and (2) SurpriseRatio (SR), a token/sequence metric used to select a small, curated subset of open-domain data to anchor the distribution and mitigate forgetting.

**Strengths:**

Clear motivation and alignment theory: The acceptance-rate analysis ties conditional MLE on teacher prefixes to reduced forward-KL and then to higher acceptance through Pinsker, which is a clean and relevant argument for SD with CoT.

**Weaknesses:**

1. **Extremely poor writing quality**: The manuscript contains **numerous obvious artifacts from LLM generation** that were never properly edited. For example, it repeatedly mixes two different types of hyphens/dashes throughout the text, creating inconsistent typography. There are also **many traces of unfinished drafts, including editorial comments left in place**. One clear example is the phrase “not clear” left at the end of line 299. In addition, **many sentences lack spaces after periods**, and **numbers are often attached directly to preceding words without spacing**. These issues make the submission appear rushed and unprofessional.

2. Unusual and weak dataset choices: Apart from GSM8K, the other two datasets used in the experiments are rarely seen in current reasoning literature. Moreover, GSM8K itself is a relatively simple dataset. It would be more convincing to evaluate on widely used and more challenging benchmarks such as AIME24, AMC, or MATH500. Since current interest in reasoning models is largely driven by long CoT performance, the paper should also specify the maximum output length used in all experiments.

3. Non-standard model selection: The chosen models, Vicuna-7B and Llama-3-8B-Instruct, are not typical reasoning models. Their inclusion weakens the persuasiveness of the results for reasoning-focused tasks.

4. Unconvincing claim about ShareGPT data: Table 2 claims that using a small amount of ShareGPT data can achieve performance comparable to EAGLE. However, the experiments are conducted only on CoT datasets, whereas EAGLE was not trained on such data. This comparison does not demonstrate the claimed effect convincingly. To support the claim, the authors should evaluate on more general-purpose benchmarks, such as MT-Bench, to verify broader capabilities.

**Questions:**

1. Can you clean up the manuscript thoroughly to remove all unfinished editorial notes, fix inconsistent dash usage, and correct spacing issues? The current presentation quality significantly detracts from the work.

2. Why did you choose such uncommon datasets for evaluation, and do you plan to include stronger reasoning benchmarks (e.g., AIME24, AMC, MATH500) with clearly stated maximum output lengths?

3. What is the rationale behind selecting Vicuna-7B and Llama-3-8B-Instruct as your experimental models, given that they are not standard reasoning models?

4. How can the claim about ShareGPT’s effectiveness be justified without evaluation on general benchmarks like MT-Bench? Would you consider adding such experiments?

---

### Official Review · Reviewer_Up4r · 2025-10-31

**Soundness:** 1
**Presentation:** 1
**Contribution:** 1
**Rating:** 0
**Confidence:** 4

**Summary:**

This work proposes a data distillation and curation approach for training draft models for improved accuracy in speculative decoding (SD) settings in which the target model outputs intermediate chain-of-thought (CoT) reasoning tokens. The proposed technique includes: 1) extracting hard CoT labels from the teacher model; 2) sample curation from open domain datasets based on the proposed SurpriseRatio metric. The proposed methodology is compared with EAGLE trained on the full ShareGPT dataset, CoT distillation only, and the mixed dataset completions only (Gold Answer).

**Strengths:**

* Training draft models for SD remains a barrier to adoption, as such distillation and data curation methods that accelerate draft model training have potential to be high impact.
*  The proposed method achieves improved speedup ratios and accepted lengths on the in-domain datasets.
* SurpriseRatio selection offers similar draft model accuracy as vanilla EAGLE with significantly less data.

**Weaknesses:**

# Major concerns
The following represent key weaknesses that must be addressed to increase the rating:
* In-domain data evaluation only: The draft models are trained on in-domain datasets with teacher CoT trajectories. It appears that the reported efficiency metrics are measured on the same in-domain datasets. Whether a train/test split was used to prevent data contamination is not made explicit. The generalizability of the proposed method to out-of-domain data is unknown. In practice, draft models are typically deployed across general tasks. Evaluating the proposed method on a more diverse dataset such as SpecBench [6] would be more indicative of real-world performance.
* Catastrophic forgetting evaluation: Empirical experiments only measure efficiency on in-domain data. Given the motivation of the work to address catastrophic forgetting, evaluating the proposed method on general-domain data is crucial to evaluating whether empirical forgetting has been mitigated.
* More robust baselines: In Table 2, EAGLE trained on the full ShareGPT dataset is compared to the proposed method with and without in-domain data and a small subset of ShareGPT data. Given that the efficiency metrics are being measured on the in-domain datasets, it would be beneficial to include a comparison to EAGLE trained on task-specific corpus such as GSM8k. EAGLE trained with the standard two-step pipeline of pretraining with ShareGPT followed by SFT with the CoT data would also help reinforce the claim made in the introduction that this simple approach is “ineffective”.
* SurpriseRatio clarity: The definition and intuition behind SurpriseRatio is confusing. What does SR measure when $x_t \notin \mathcal{N}_k$? SR is claimed to compare “the draft model’s surprisal with that of the target model”; however, the target logit distribution is not included in the metric.
* Reproducibility: Several hyperparameters required to reproduce the work are missing. I.e.., learning rate. L312 notes that the sampling hyperparameters are inherited from EAGLE but it’s unclear if this extends to the training hyperparameters too.
* Lack of novelty in sequence distillation approach: Using hard labels generated by the teacher models to improve draft/target model alignment has been previously explored [1-5]. The extension of KD to CoT traces does not represent a significant extension of these prior methods in my opinion. Adding these works to the knowledge distillation related work section would improve the paper.
* Proofreading: Numerous typos are present in the paper, see below for a full list.

# Minor concerns
The following are minor concerns, typos, etc. which would improve the work but do not affect the final rating:
* L024: “by2.76x”
* L033: "eliciting the model to and generate…”
* L049: “can be (a) two-step pipeline”
* L50: (SFT).Unforunately, ...”
* L192: “tearcher”
* Figure 1: Based on the draft model logits, should the Low SR token be [to]? How is SR defined for tokens other than $x_t$?
* L280: “Eq. equation”
* L299: “(Not Clear)”
* L329: “Settings as eagle” -> “settings as EAGLE”
* L334: Appendix B is not included in the SI
* Formal definition of “Gold Answer” in Tables is missing.
* L377: “Figure 2,ground”
* L379: “targets.By”
* L406: “speedUp ratio”
* L430: “traces.By”
* L460: “draft,highlighting”

[1] R. Agarwal et al., “O1n-Policy Distillation of Language Models: Learning from Self-Generated Mistakes,” Jan. 17, 2024, arXiv: arXiv:2306.13649. doi: 10.48550/arXiv.2306.13649.

[2] W. Xu et al., “Speculative Knowledge Distillation: Bridging the Teacher-Student Gap Through Interleaved Sampling,” Oct. 15, 2024, arXiv: arXiv:2410.11325. doi: 10.48550/arXiv.2410.11325.

[3] X. Liu et al., “Online Speculative Decoding,” Jun. 10, 2024, arXiv: arXiv:2310.07177. doi: 10.48550/arXiv.2310.07177.

[4] R. Goel, M. Gagrani, W. Jeon, J. Park, M. Lee, and C. Lott, “Direct Alignment of Draft Model for Speculative Decoding with Chat-Fine-Tuned LLMs,” May 13, 2024, arXiv: arXiv:2403.00858. doi: 10.48550/arXiv.2403.00858.

[5] V. Thangarasa, G. Venkatesh, M. Lasby, N. Sinnadurai, and S. Lie, “Self-Data Distillation for Recovering Quality in Pruned Large Language Models,” presented at the Eighth Conference on Machine Learning and Systems, Feb. 2025. Available: https://openreview.net/forum?id=ewkcZuU9Gk

[6] H. Xia et al., “Unlocking Efficiency in Large Language Model Inference: A Comprehensive Survey of Speculative Decoding,” Jun. 04, 2024, arXiv: arXiv:2401.07851. doi: 10.48550/arXiv.2401.07851.

**Questions:**

* How does the proposed method perform on out-of-domain data such as SpecBench?
* Was a train/test split used for evaluating on GSM8k and the other in-domain datasets?
* How does EAGLE pretrained on two-step pipeline of ShareGPT pretraining and SFT on CoT traces perform compared to the proposed method?
* SurpriseRatio selects samples where the draft/target are already well-aligned. What about the inverse? If we instead select data where $q(x_t | x_{<t})$ and $p(x_t | x_{<t}) greatly differ, does the proposed method have worse acceptance length?

---

### Official Review · Reviewer_zAsQ · 2025-11-01

**Soundness:** 4
**Presentation:** 2
**Contribution:** 3
**Rating:** 4
**Confidence:** 4

**Summary:**

This paper addresses the problem of catastrophic forgetting in speculative decoding methods that require the draft model to be aligned with the large model via distillation, specifically for CoT trajectories. Motivated by previous insights that mixed training with open-domain data helps against catastrophic forgetting, they propose distilation over a mix of CoT trajectories (following previous work) and open-domain data.

A key contribution is that the open-domain data are filtered such that the draft and large model exhibit the same level of surprisal over each sequence's tokens, measured as the probability of the teacher-forced token over the probability mass of the draft model's top-k tokens for that time-step.

The paper also formally establishes the inverse relation between forward-KL convergence and acceptance rate, theoretically supporting its methodology.

The paper finally conducts experimentation and ablation analysis over two models and three datasets, showing that the mixed training can achieve significant speedups.

**Strengths:**

- The surprisal-based metric for open domain data filtering is novel and shown to be effective through ablation studies.
- The experimental section shows consistent speed gains using the proposed mixed training.

**Weaknesses:**

- It is unclear why the proposed training paradigm is applicable only on CoT settings. The paper should also consider standard SD settings, and compare against more recent SD methods.
- The purpose of the surprisal-based filter is a bit unclear in the paper; see questions.
- A minor weakness but the paper could benefit from some better proofreading to improve its presentation, including missing spaces, inconsistent capitalization, and numerous typos.

**Questions:**

- The surprisal-based filter is proposed as matching the surprisal levels between draft and large model, but equation 5 does not consider the surprisal of the large model.
- How is the value of T_SR in eq. 6 established? It was unclear from the paper.
- In Table 2, why is the perplexity baseline missing from the ALPACA setting?
- In Section 5.2, the paper claims to compare "three levels of supervision", but Figure 2 only contains two. Please clarify this.
- I am afraid that Table 2 is rather confusing. Unsure why the data mixtures are described as metrics, as well as how the comparison against EAGLE should be framed. Please elaborate on this.
- Table 1 and 3 may be placed in the wrong sections. The paper never directly refers to them.

---

### Official Review · Reviewer_MakW · 2025-11-01

**Soundness:** 2
**Presentation:** 2
**Contribution:** 2
**Rating:** 4
**Confidence:** 3

**Summary:**

This paper addresses the inefficiency of speculative decoding (SD) under Chain-of-Thought (CoT) prompting, which suffers from catastrophic forgetting and distribution shift in training-based SD methods. The authors propose a CoT-aware mixed training framework that combines process-level CoT distillation with feature regression to reduce distribution divergence and a SurpriseRatio (SR)-guided data selection metric to anchor distribution and prevent forgetting. A two-stage mixed-training schedule balances task alignment and generalization.

**Strengths:**

1. It attempts to address the core pain points of speculative decoding (SD) under Chain-of-Thought (CoT) prompting—catastrophic forgetting and distribution shift in training-based SD methods. The proposed mixed framework combining "process-level CoT distillation" and "SR-guided data selection" targets a specific gap in existing research

2. Basic mathematical derivations are provided, such as the equivalence between NLL and KL divergence, and the connection between acceptance rate and Total Variation Distance

**Weaknesses:**

1. The SurpriseRatio (SR), a key component of the framework for open-domain data selection, lacks clarity in critical details. The rationale for choosing the value of k, the aggregation method for sequence scores, and the range of threshold tuning are all unspecified. The paper only mentions "tuning on a held-out split" without providing concrete information

2. Experiments are restricted to mathematical reasoning tasks, lacking performance on general tasks, even though the paper claims they solved the forgetting problem through a data mixing strategy.

3. Comparisons are only made against vanilla EAGLE and simple variants (CoT distillation and golden answer). Meanwhile, in table 3, it can be seen that most of the acceleration comes from CoT distillation.

4. Although the paper derives theoretical relationships (e.g., "reducing forward KL lowers TVD and increases acceptance rate") , it does not empirically verify these assumptions. Thus, the causal link between the proposed theoretical mechanism and performance gains remains unproven.

**Questions:**

1. In abstract, this paper claim that "our methods achieve wall-clock speedups of 3.04×–4.55× and average acceptance length by 2.76×–5.62×". The speedup is compared to vanilla decoding method, but the average acceptance length is compared to which model (vanilla model can not compute average acceptance length)?

2. Can the proposed framework be extended to reasoning model (eg. Qwen3-8B), which does not need specific prompts to elicit CoT ?

---

### Meta-Review · Area_Chair_ZjRu · 2026-01-03

**Summary:**

This paper studies speculative decoding (SD) under CoT prompting and identifies catastrophic forgetting and distribution shift as key challenges when training draft models. To address this, this work proposes a data distillation and curation approach for training draft models, which includes (i) process-level CoT distillation with feature regression and (ii) a SurpriseRatio (SR)-guided data selection strategy to curate open-domain data that anchors the draft model distribution. The paper also provides a theoretical argument linking forward-KL minimization to higher acceptance rates and reports wall-clock speedups and acceptance length gains on several reasoning datasets.

Overall, reviewers agree that catastrophic forgetting in SD draft-model training, particularly with CoT trajectories, is a real and important issue. The reported speedups and acceptance-length improvements might benefit future work if it can generalize to out-of-domain evaluations. However, there are several significant concerns regarding:

- experimental setting (only in-domain, unfair comparison with baseline EAGLE trained on the full ShareGPT dataset, and so on)
- justification of SurpriseRatio (like key hyperparameters’ definition and selection, connections between theoretical claims and experiments are not explicit)
- empirical validation of the proposed approach: uncommon datasets and models, unfair comparison with baselines
- novelty and poor writing quality (e.g., many typos, many traces of unfinished drafts, including editorial comments left in place, numerous obvious artifacts from LLM generations and so on)
See details in concerns.

**Reviewer Concerns:**

1. Experimental scope and generalization

- Experiments are largely limited to in-domain reasoning datasets, primarily GSM8K and two less common benchmarks.
- There is no convincing evaluation on general-domain or out-of-domain benchmarks (e.g., SpecBench, MT-Bench), despite claims about mitigating catastrophic forgetting and maintaining generality.
- It is unclear whether strict train/test separation was enforced, raising concerns about potential data contamination.

2. SurpriseRatio clarity and validation

- Across all reviews, SR is consistently flagged as under-specified and confusing. Key hyperparameters (e.g., choice of k, aggregation across tokens, thresholds) are not clearly defined.
- The metric is claimed to match draft and target surprisal, but the target model distribution does not explicitly appear in the formulation.
- Theoretical claims about distribution anchoring and forgetting mitigation are not directly validated empirically.

3. Extremely poor writing

- All reviewers criticize the writing quality. The manuscript contains numerous typos, formatting issues, inconsistent notation, and even leftover editorial comments, which significantly detract from clarity and professionalism.
- Several tables and figures are poorly explained, misplaced, or internally inconsistent.


4. Experiments setups, baseline comparisons and novelty

- Comparisons are mostly against vanilla EAGLE and simple variants. Stronger and more fair baselines (e.g., EAGLE trained with task-specific data, two-step ShareGPT + SFT pipelines, or more recent SD methods) are missing.
- The choice of Vicuna-7B and Llama-3-8B-Instruct, which are not standard reasoning models, weakens the paper’s relevance to current reasoning-focused SD research.
- More challenging and widely used reasoning benchmarks (e.g., AIME24, AMC, MATH500) are absent, and maximum output lengths are not clearly specified.
- Several reviewers questioned the novelty of extending knowledge distillation to CoT traces, less novelty.

**Reviewer Scores:**

2

---

### Decision · Program_Chairs · 2026-01-26

Reject